# Crack Detection of Concrete Based on Improved CenterNet Model

Huaiqiang Kang [1], Fengjun Zhou [1], Shen Gao [1] and Qizhi Xu [2],*

[1] Defence Engineering Institute, AMS, PLA, Beijing 100850, China
[2] School of Mechatronical Engineering, Beijing Institute of Technology, Beijing 100081, China
* Correspondence: qizhi@bit.edu.cn

**Abstract:** Cracks on concrete surfaces are vital factors affecting construction safety. Accurate and efficient crack detection can prevent safety-related accidents. Using drones to photograph cracks on a concrete surface and detect them through computer vision technology has the advantages of accurate target recognition, simple practical operation, and low cost. To solve this problem, an improved CenterNet concrete crack-detection model is proposed. Firstly, a channel-space attention mechanism is added to the original model to enhance the ability of the convolution neural network to pay attention to the image. Secondly, a feature selection module is introduced to scale the feature map in the downsampling stage to a uniform size and combine it in the channel dimension. In the upsampling stage, the feature selection module adaptively selects the combined features and fuses them with the output features of the upsampling. Finally, the target size loss is optimized from a Smooth L1 Loss to IoU Loss to lessen its inability to adapt to targets of different sizes. The experimental results show that the improved CenterNet model reduces the FPS by 123.7 Hz, increases the GPU memory by 62 MB, increases the FLOPs by 3.81 times per second, and increases the AP by 15.4% compared with the original model. The GPU memory occupancy remained stable during the training process and exhibited good real-time performance and robustness.

**Keywords:** crack detection; attention mechanism; feature fusion; frameless; CenterNet

## 1. Introduction

With the rapid development of China's economy, civil engineering construction projects are increasing, and as one of the pillar industries of the economy, the construction industry has played an irreplaceable role in national construction. With the increasing number of buildings, roads, bridges, tunnels, and other infrastructures, maintaining them in good working condition is extremely important for public safety. Concrete cracks are usually caused by internal stress and environmental action, leading to the internal fatigue of the material and resulting in cracks and fractures on the surface of the concrete [1]. The occurrence of cracks often represents a change in the structure where the cracks occur. Over time, further cracking and falling off often occur, and water seepage occurs. Therefore, crack detection is of great significance for the healthy operation of construction projects [2]. Based on the location of cracks in the material, they can be divided into surface and internal cracks. The main research object of this study was the surface cracks in construction engineering concrete.

Surface crack detection methods include eye observation, ultrasonic detection [3], eddy current detection [4], speckle interference [5], penetration detection [6], laser holography [7], X-ray detection [8], and computer vision detection [9]. Most of the aforementioned methods have formed a relatively complete detection system that can perform surface crack detection well; however, they also have their adaptation scenarios and shortcomings. For example, although ultrasonic detection is sensitive to planar defects, it is difficult to detect nonplanar structures owing to acoustic coupling, and the surface crack detection effect of arched structures facing some projects could be better. Although the detection accuracy is high, optical detection is significantly affected by ambient light interference and vibrations during

actual operation. The infrared detection method has a fast detection speed; however, the detection environment is limited due to the equipment's large size. Current computer vision detection technology often obtains the surface image or video of the research object through a camera and other sensing equipment; then, the obtained image or video is pre-processed and feature extracted, and different algorithm models are trained and tested to finally achieve the purpose of target recognition or positioning [10].

In the use of deep learning methods to deal with crack detection problems, researchers generally used image processing algorithms to make predictions. Wang Fan [11] studied the problem of crack detection using a mathematical morphology and image fusion. Chambon [12] conducted a study of road crack detection and evaluation using computer vision. Tongji University studied an MTI-100 tunnel-detection system and achieved crack detection and location [13]. Soukup [14] used convolutional neural networks to detect the surface cracks. Using an adaptive iterative method, Peng [15] used an improved Otsu threshold segmentation algorithm to study crack images. Yang [16] proposed a new image analysis method for concrete crack detection and conducted a detailed study of a detection method based on edge cracks. Cao Jianbing [17] completed the study of visual feature maps and proved that the neural network can effectively distinguish the difference between the features of split and non-crack images, which led to significant progress. Sun Yunpeng [18] conducted an analysis of existing deep learning object detection algorithms, obtained the structure and design method of the crack detection network based on convolutional neural network, and realized the image crack detection based on DeepLabV3+. Shao Jiang [19] proposed an automatic crack detection method for concrete Bridges based on machine vision, which realized the automatic identification and measurement of cracks and improved the detection efficiency. Fernandez [20] studied a decision-tree heuristic algorithm for crack detection and achieved satisfactory simulation results. Guan Shijie [21] applied the SegNet network to surface crack detection and achieved satisfactory results. Zhao Xuefeng [22] proposed a concrete crack detection model based on artificial intelligence and smart phones and used a convolutional neural network in artificial intelligence deep learning to identify and locate cracks in pictures, achieving the purpose of crack detection. Li [23] studied a concrete surface crack detection method by combining the improved active contour model with the Canny iterative operator; however, the operation time was relatively long, and there were certain limitations. Yang Song [24] used the Canny operator to extract the crack edge and the automatic generation technology to generate the finite element mesh of the crack structure to realize the analysis of the stress in the crack region.

In the field of combining drone aerial photography technology and computer vision technology, many scholars have studied it. Pan Xiang [25] conducted ground target detection using UAV and computer vision technology. Zhang Ruixin [26] proposed a method in order to obtain more accurate detection boxes, in which the label encoding strategy and bounding box regression method of CenterNet were optimized, introducing localization quality loss to improve the localization quality of the detection boxes. With the continuous development of computer vision technology and UAV aerial photography technology, the images of concrete cracks are being collected by unmanned aerial vehicles and then processed as data. Finally, the cracks are identified by computer vision technology. The method has the advantages of simple principle, convenient operation, strong flexibility, high precision, low cost, and no contact. This article utilizes computer vision technology to process images collected by drones and achieves good results, which have been applied in practical engineering to a certain extent.

## 2. CenterNet

The CenterNet algorithm is a single-stage model without an anchor frame that was first proposed in 2019 [27]. The CenterNet algorithm has the characteristics of high precision, fast training speed, and a simple network structure. The principle of the CenterNet model is as follows: the center point of the target is used to replace the anchor frame, the peak value of the thermal map is used as the center point of the detection object, the threshold is

then set for screening and comparison of the target center point, and, finally, the category information is obtained via regression using the image features [28]. The training process of CenterNet does not need to consider the anchor mechanism, nor does it need to set or postprocess hyperparameters in advance, significantly reducing the computational load on the entire network [29].

The original CenterNet uses ResNet18, DLA-34, and Hourglass convolutional networks for feature extraction and then transfers the feature map to the detection module for processing. Finally, the target centre point and category, target length and width prediction, and centre point bias are transferred through the convolution operation. A schematic of the CenterNet algorithm is shown in Figure 1.

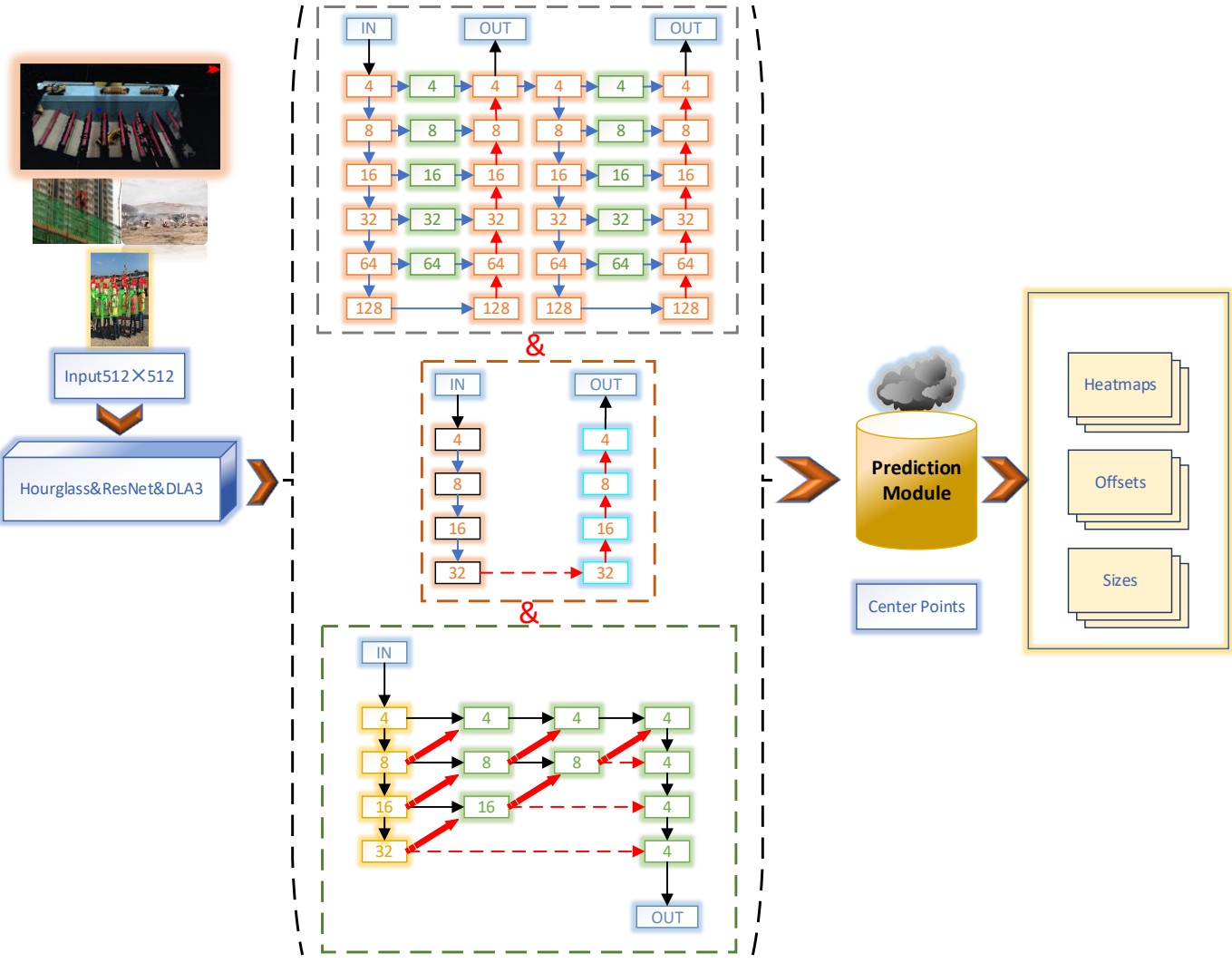

**Figure 1.** Schematic diagram of CenterNet algorithm model [30].

The CenterNet algorithm model makes predictions through three convolution blocks: target centre point and category, target length and width prediction, and centre point bias. The loss function of the CenterNet algorithm consists of the loss function of the centre point and classification, loss function of the target frame size, and loss function of the centre point bias [31].

The loss function $L_k$ of the centre point and classification is the focal loss function, and the calculation formula is shown in Equation (1) [32].

$$L_k = -\frac{1}{N}\sum_{xyc}\begin{cases} (1 - \hat{Y}_{xyc})^\alpha lg\ (\hat{Y}_{xyc}),\ Y_{xyc} = 1 \\ (1 - \hat{Y}_{xyc})^\beta (\hat{Y}_{xyc})^\alpha \times lg\ (g - \hat{Y}_{xyc}), \\ others \end{cases} \tag{1}$$

In the above formula, the subscript k in the centre point and classification loss function $L_k$ represent the kth input image, $N$ represents the number of keypoints in the image, subscript *xyc* represents the positive and negative samples of the image, and $Y_{xyc}$ represents the label of the true value.

The centre point bias loss function $L_{offset}$ adopts the $L_{loss}$ function, and the calculation formula is shown in Equation (2) [33]:

$$L_{offset} = \frac{1}{N}\sum_p \left| \hat{O}_{\tilde{p}} - (\frac{p}{R} - \tilde{P}) \right| \tag{2}$$

In the above formula, *P* is the coordinate of the true value of the original image target, and R is the subsampling multiple.

The $L_{loss}$ function is used for the target frame size loss function $L_{size}$. The calculation formulas are shown in Equation (3) [34], where $S_k$ represents the size of the original target frame.

$$L_{size} = \frac{1}{N}\sum_{k=1}^N \left| \hat{S}_{pk} - S_k \right| \tag{3}$$

The final loss function was obtained by multiplying the loss function of the centre point by the classification, the loss function of the target frame size, and the loss function of the centre point bias by the corresponding coefficients, as shown in Equation (4). $L_k$ represents the central point loss function, $L_{size}$ represents the the size loss function, and $L_{off}$ represents the offset loss function; the setup is $\lambda_{size}$ = 0.1 and $\lambda_{off}$ = 1.

$$L = L_k + \lambda_{size}L_{size} + \lambda_{off}L_{off} \tag{4}$$

## 3. CenterNet Optimization

The improvement of the original CenterNet model includes three aspects: adding a new channel space attention mechanism, adding a feature selection module, and optimizing the loss function.

### 3.1. Addition of Channel Space Attention Mechanism

In the convolutional block attention module (CBAM), the channel attention uses global average pooling and global maximum pooling to obtain the global statistics for each channel, and it learns the weight of the channel through two fully connected layers. Each channel was scaled using a sigmoid function to normalize the weights between 0 and 1. Finally, the scaled channel features were multiplied by the original features to produce features with enhanced channel importance [35,36].

The function of the channel attention mechanism was to continuously enhance the importance of the channel during the training process to improve the training effect on the network. The attention mechanism diagram of the CenterNet channel used in this study is shown in Figure 2.

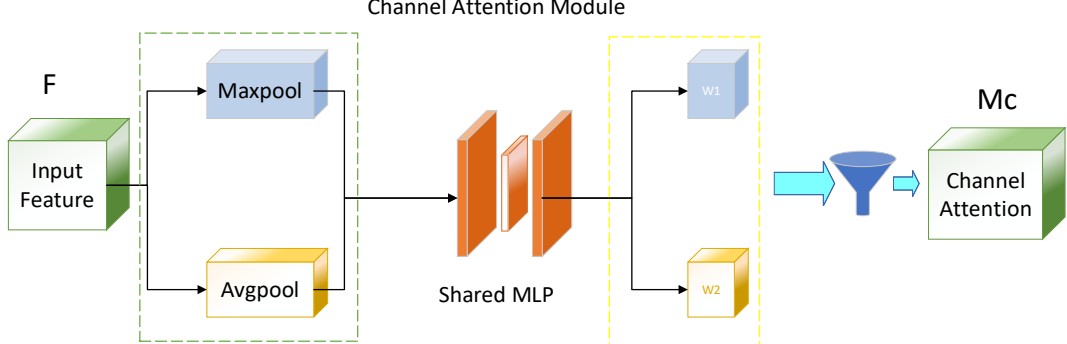

**Figure 2.** CenterNet channel attention mechanism diagram.

The spatial attention module in the CBAM uses maximum and average pooling to obtain the maximum and average values for each spatial position. As more channels were generated after convolution, the function of the spatial attention mechanism was to perform maximum pooling and average pooling operations on the channels of each feature point, obtain two different results, concatenate them, and then learn the weight of each spatial position through a convolution layer and sigmoid function. Finally, weights were applied to each spatial position on the feature map to produce features with enhanced spatial importance.

By introducing an attention module, the spatial attention mechanism enabled the model to learn the attention weights of different regions adaptively so that it could pay more attention to important image regions while ignoring unimportant ones [37]. The spatial attention mechanism of CenterNet added in this study is shown in Figure 3.

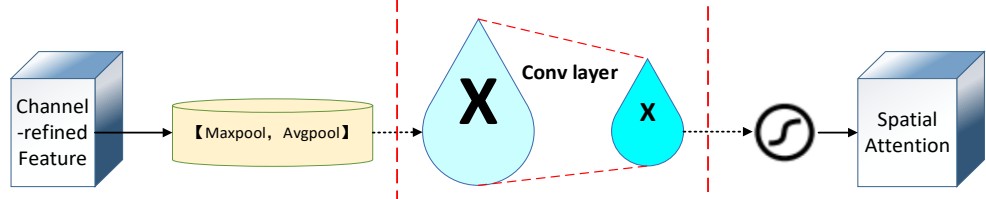

**Figure 3.** CenterNet schematic diagram of spatial attention mechanism.

In this study, a channel space attention mechanism was added, and a model combining channel and space attention was constructed to enhance the focus of convolutional neural networks on images and improve the algorithm's performance. The original network joining the CBAM mechanism is illustrated in Figure 4.

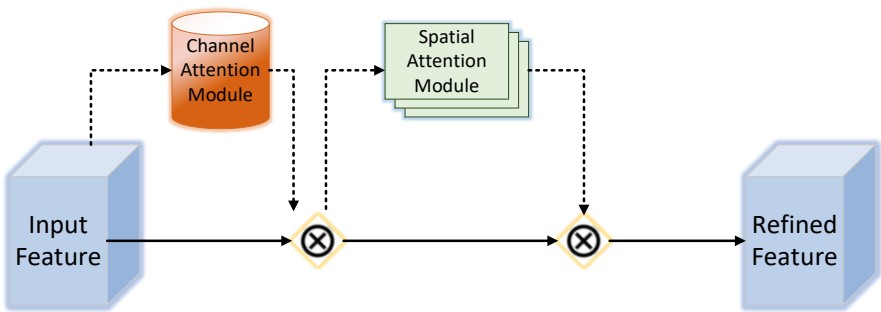

**Figure 4.** CenterNet overall CBAM mechanism schematic.

### 3.2. Addition of Feature Selection Module

After adding the feature selection module, the feature map in the downsampling stage was scaled to a unified size and combined with the channel dimensions. In the upsampling stage, the feature selection module adaptively selected the combined features and then added them to the output features. The structure of the feature selection module is illustrated in Figure 5.

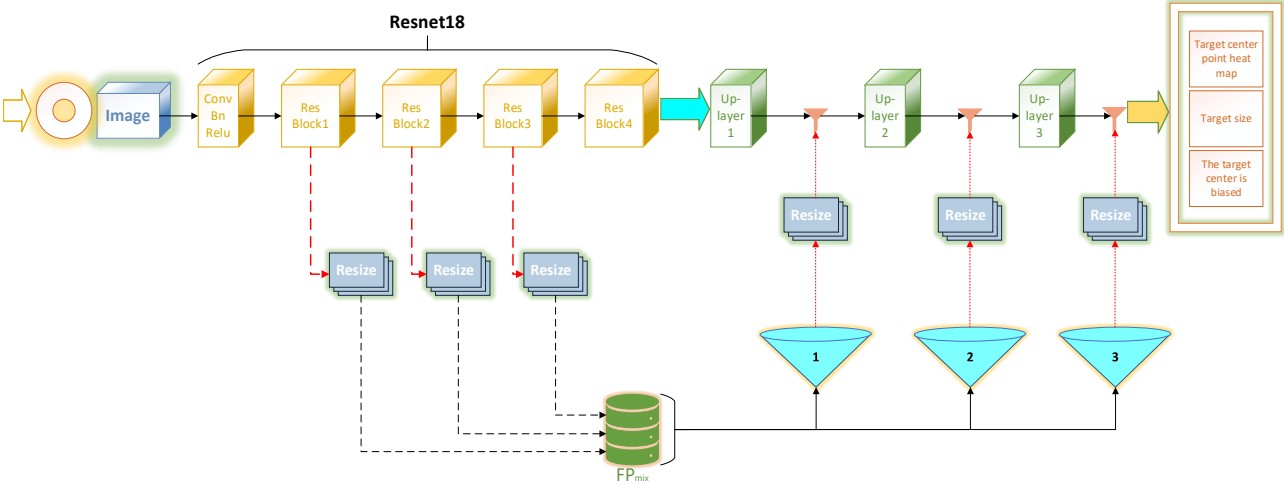

**Figure 5.** Feature selection module diagram.

When the CenterNet model was proposed, the original network only extracted the most profound feature map for detection, which led to the poor retention of deep and shallow semantic information in the entire network during training, ultimately leading to a decline in the accuracy of the entire network. The feature selection module added in this study effectively enhanced the network extraction of target features and had a stronger ability to capture effective features. The details of the feature selection module are shown in Figure 6.

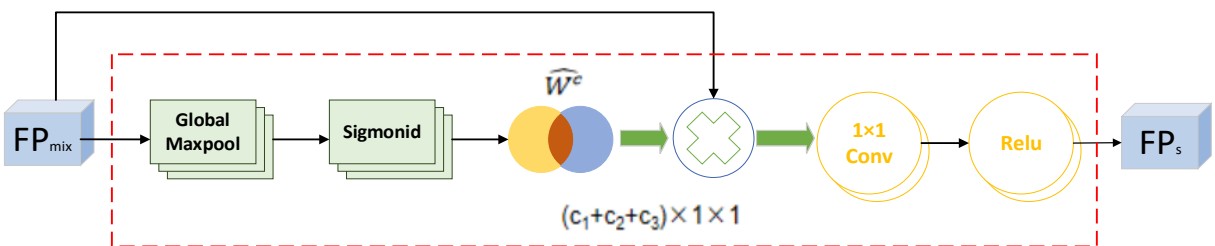

**Figure 6.** Feature selection module adds detail.

### 3.3. Optimization of the Loss Function

The target size loss changed from Smooth L1 Loss to *IoU* Loss because Smooth L1 Loss could not adapt to targets of different sizes. The calculation formula is shown in Equation (5). When calculating the *IoU* Loss, it was assumed that the centre point is the same, and the calculation formula is shown in Equation (6):

$$L_{Iou} = 1 - \frac{|A \cap B|}{|A \cup B|} \tag{5}$$

$$L_{IOU} = \ln(IOU(box_1, box_2)) \tag{6}$$

The *IoU* Loss is an indicator used to evaluate the distance between two rectangular boxes. This indicator has all the distance characteristics, including symmetry, nonnega-

tivity, identity, and triangular inequality. The advantages of using the *IOU* Loss include the following:

1. It can more accurately measure the matching degree between the prediction box and the real box.
2. It has scale invariance, which means that regardless of the sizes of the prediction box and the actual box, as long as they are located near each other, their IoU values will be similar. This helps the model to have a better generalization ability when dealing with objects of different scales and sizes.

### 4. Experiment and Result Analysis

#### 4.1. Experimental Environment

The experimental environment used in this paper was an Ubuntu18.04 64-bit operating system with 754 GB running memory, a Tesla V100S graphics card with 32 GB graphics memory, and an Intel(R) Xeon(R) Gold 6240 CPU. The PyTorch deep learning framework was used to build the model with CUDA version 10.1 and cudnn version 7.6.0.

#### 4.2. Evaluation Index

Generally, current mainstream computer vision algorithm model evaluation indicators include accuracy and performance [38]. The index used to measure the accuracy of the target detection algorithm is generally the AP, and the performance index includes the FLOPs, FPS, and video memory occupations. The evaluation indicators are shown in Table 1.

**Table 1.** Evaluation index and meaning interpretation.

| Index | Implication |
|---|---|
| FLOPs | The number of floating-point operations used to measure the computational complexity of the model |
| FPS | The number of images the algorithm processes per second—the higher the value, the faster the algorithm processes |
| p | The size of the video memory occupied by the algorithm in the inference stage—the smaller the video memory occupation, the fewer resources are required |

Average Precision (*AP*) was obtained by calculating the area of the PR curve. The calculation formula is shown in Equation (7) [39]:

$$AP = \int_0^1 p(\tau)d(\tau) \tag{7}$$

#### 4.3. Data

The experimental dataset in this study consisted of 3000 crack pictures captured via the UAV, which were divided into training and test sets at a 9:1 ratio.

In the pre-processing stage, part of the training set was augmented to improve the generalization ability of the algorithm model. The image transformation method adopted in the dataset enhancement was still close to the tunnel crack image collected after image processing, including random brightness transformation, random horizontal flipping, and random vertical flipping. The transformation results after processing are shown in Figure 7.

The image was scaled and standardized before being input into the network. The widths and heights of the scaled images were 512. The mean values of the standardized RGB three-channel were 123.675, 116.28, and 103.53, and the standard deviation was 58.395, 57.12, 57.375.

CenterNet determines the target's location by predicting the target centre point, target centre point bias, and target size. Therefore, the corresponding labels of the image include the target centre-point Gaussian heat map, target centre-point bias, and target size, which are represented by a tensor of the same size as the network output.

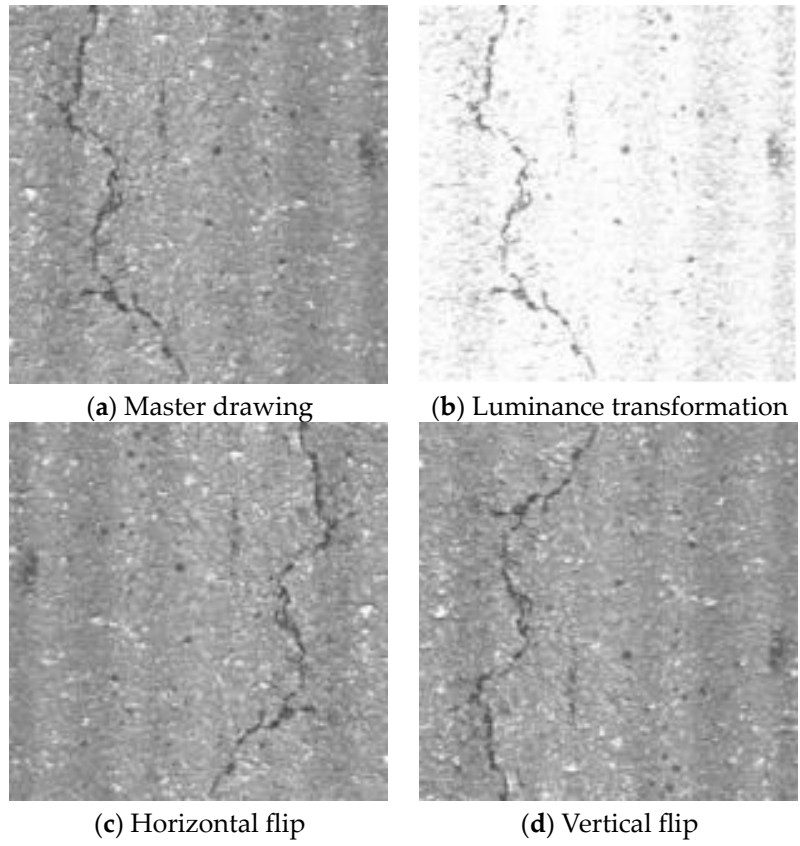

(**a**) Master drawing        (**b**) Luminance transformation

(**c**) Horizontal flip        (**d**) Vertical flip

**Figure 7.** Random transformation used in training.

### 4.4. Training Process and Experimental Results

To ensure the real and effective results of the comparative experiments, the training parameters used in all the experiments involved in this study were completely consistent. The initial learning rate of the training was 0.0001. The cosine annealing learning rate adjustment method was adopted, and the minimum learning rate was 0.00001. The batch size was set to eight during the training process. A total of 300 epochs were trained using the SGD optimization algorithm.

The training experiments were conducted in five groups: original CenterNet with the backbone network of ResNet18, CenterNet with the channel space attention mechanism, CenterNet with the feature selection module, CenterNet with target size loss improvement, and CenterNet with the above three improvements. Table 2 compares the performance of CenterNet with the addition of CBAM and feature-selection modules, including FLOPS, FPS, and video memory.

**Table 2.** Comparison of network performance before and after CenterNet optimization.

| Network | FLOPs | Memory Footprint/MB | FPS | Video Memory/MB |
|---|---|---|---|---|
| CenterNet | 13.06 | 50.3 | 296.5 | 1347 |
| CenterNet-CBAM | 13.06 | 51.7 | 189.9 | 1349 |
| CenterNet-FS | 16.35 | 51.1 | 250.2 | 1405 |

In the data training process, owing to the different difficulties involved in data feature extraction, there are overlaps and omissions in some data, as shown in Figure 8. Given this situation, the optimized model used in this study adopts the method of strengthening the feature extraction. This situation changed significantly after adding the feature extraction module, and the data processing accuracy was effectively improved.

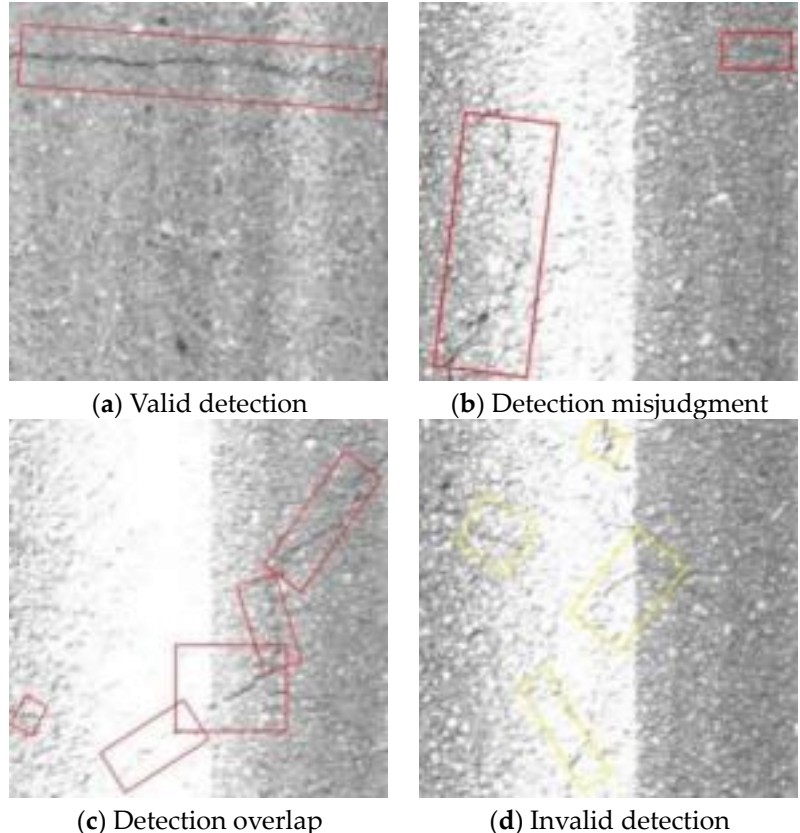

(**a**) Valid detection　　　　　(**b**) Detection misjudgment

(**c**) Detection overlap　　　　　(**d**) Invalid detection

**Figure 8.** Detection detail.

When the test environment of the controlled experiment was the same as that of the training environment, the batch size of the experiment was set to one. The ablation experiments are summarized in Table 3. From the ablation experiment, the following results were obtained:

**Table 3.** CenterNet optimized process ablation experiment.

| Serial Number | CenterNet | CBAM | FS | Iou | Memory Footprint/MB | FPS | Video Memory/MB | FLOPs | AP |
|---|---|---|---|---|---|---|---|---|---|
| 1 | √ | × | × | × | 50.3 | 296.5 | 1347 | 13.06 | 0.751 |
| 2 | √ | √ | × | × | 51.7 | 189.9 | 1349 | 13.06 | 0.823 |
| 3 | √ | × | √ | × | 51.1 | 250.2 | 1405 | 16.35 | 0.852 |
| 4 | √ | × | × | √ | 50.8 | 172.8 | 1378 | 15.26 | 0.772 |
| 5 | √ | √ | √ | × | 52.3 | 237.1 | 1382 | 15.43 | 0.864 |
| 6 | √ | √ | × | √ | 51.8 | 196.8 | 1375 | 16.49 | 0.872 |
| 7 | √ | × | √ | √ | 51.9 | 253.7 | 1401 | 16.02 | 0.869 |
| 8 | √ | √ | √ | √ | 52.4 | 270.9 | 1409 | 16.87 | 0.905 |

After the CBAM module was added, the model size increased by 1.4 MB, FPS decreased by 106.6, video memory increased by 2 MB, FLOPs remained unchanged, and AP increased by 0.072 compared with the original model.

After adding the feature selection module, the model size increased by 0.8 MB, FPS decreased by 46.3, video memory increased by 58 MB, FLOPs increased by 3.29, and AP increased by 0.101 compared with the original model.

After IOU optimization in the original model, the size increased by 0.5 MB, FPS decreased by 123.7, video memory increased by 31 MB, FLOPs increased by 2.2, and AP increased by 0.021 compared with the original model.

After the addition of the feature selection module, the optimized model decreased the target size loss faster than the original CenterNet because the feature selection module could adaptively select the underlying features (such as the target texture and edge information) in the downsampling process and add them to the feature map in the upsampling process. Thus, the target size could be learned more quickly.

The change curve for the CenterNet target size loss after the original CenterNet and the addition of the feature selection module are shown in Figure 9.

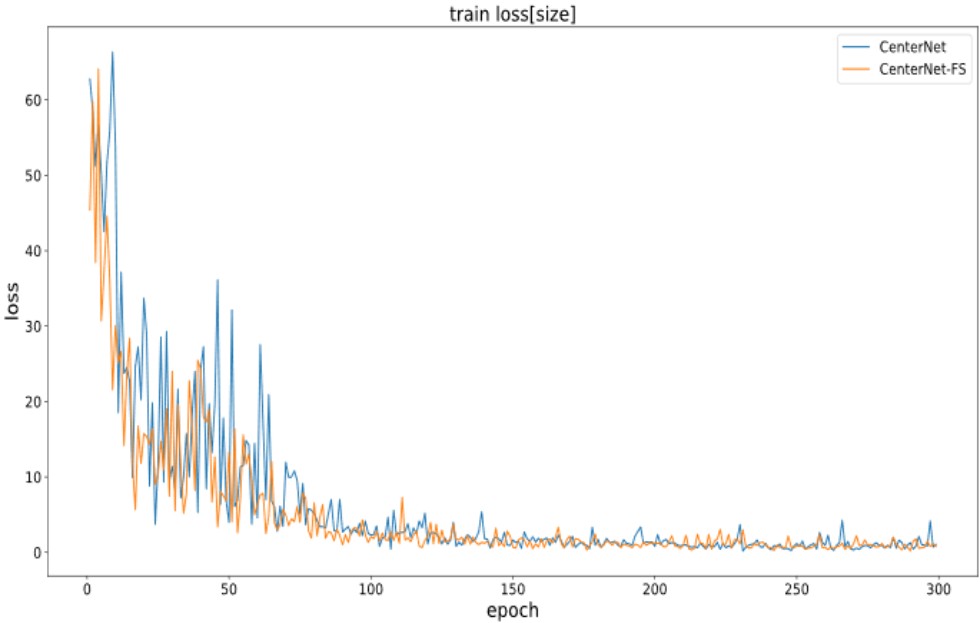

**Figure 9.** Loss curve of network target size between CenterNet and feature selection module.

The feature selection module can adapt to underlying features, which is also evident in the actual detection effect. As shown in Figure 10, after adding the feature selection module, the optimized model can predict the crack size more accurately due to the inclusion of information such as the crack edge.

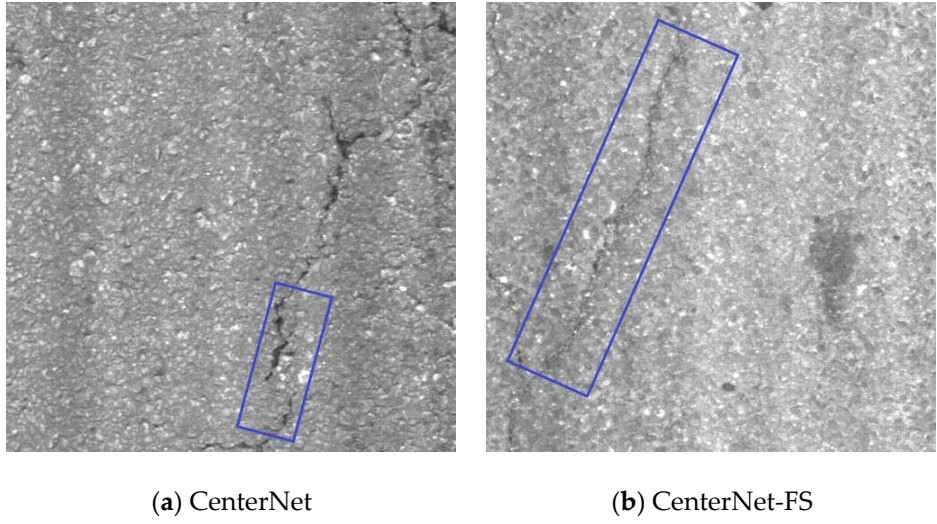

(**a**) CenterNet          (**b**) CenterNet-FS

**Figure 10.** Comparison of detection results between CenterNet and feature selection module.

The CBAM and feature selection modules, particularly the CBAM module, significantly impact the reasoning speed of the network. This is because, after the CBAM module is added to each ResBlock, the FPS of the network decreases overall, whereas the feature

selection module reduces the FPS. Regarding video memory usage, the impact of the two additional modules was relatively small.

The feature information of the entire network is compressed by the subsampling module, which reduces the workload of subsequent network training and increases the reasoning speed of the entire network. The input information in the upper layer of the network is enhanced after the feature extraction module, and the upsampling stage uses fewer convolutional layers to improve the running speed of the network. The information about each input and output layer of the overall network optimized in this study is shown in Table 4.

**Table 4.** CenterNet improved network layer input and output.

| Net | Input Size | Input Channel | Output Size | Output Channel |
|---|---|---|---|---|
| Convolution 1 | $512 \times 512$ | 3 | $128 \times 128$ | 64 |
| Res-Block1 | $128 \times 128$ | 64 | $128 \times 128$ | 64 |
| CBAM1 | $128 \times 128$ | 64 | $128 \times 128$ | 64 |
| Res-Block2 | $128 \times 128$ | 64 | $64 \times 64$ | 128 |
| CBAM2 | $64 \times 64$ | 128 | $64 \times 64$ | 128 |
| Res-Block3 | $64 \times 64$ | 128 | $32 \times 32$ | 256 |
| CBAM3 | $32 \times 32$ | 256 | $32 \times 32$ | 256 |
| Res-Block4 | $32 \times 32$ | 256 | $16 \times 16$ | 512 |
| CBAM4 | $16 \times 16$ | 512 | $16 \times 16$ | 512 |
| Upper sampling layer 1 | $16 \times 16$ | 512 | $32 \times 32$ | 256 |
| Upper sampling layer 2 | $32 \times 32$ | 256 | $64 \times 64$ | 128 |
| Upper sampling layer 3 | $64 \times 64$ | 128 | $128 \times 128$ | 64 |
| Target center point | $128 \times 128$ | 64 | $128 \times 128$ | 1 |
| The target center is biased | $128 \times 128$ | 64 | $128 \times 128$ | 2 |
| Target size | $128 \times 128$ | 64 | $128 \times 128$ | 2 |

To demonstrate the improvement in the performance of the model before and after optimization more intuitively, five groups of training processes were randomly selected for comparison, as shown in Figure 11. Dark blue represents the data processing accuracy of the original CenterNet model, and yellow represents the improvement in accuracy brought about by the optimized CenterNet-CBAM-FS-IOU model.

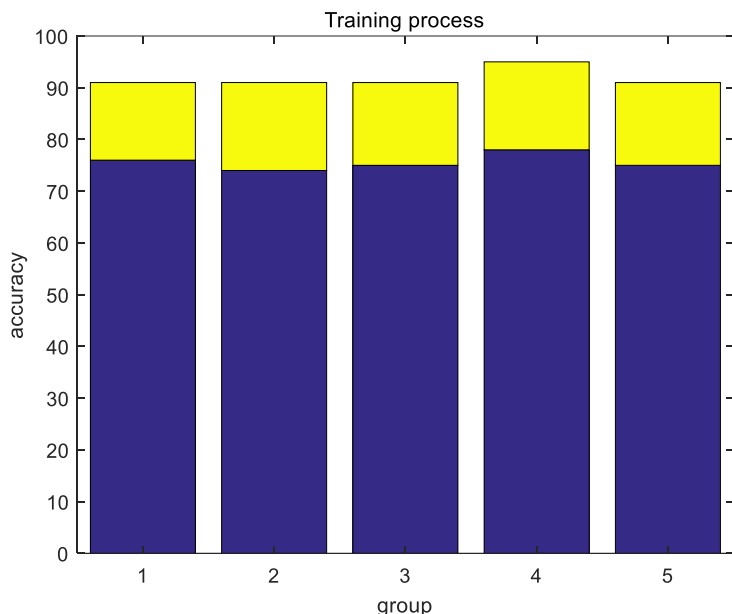

**Figure 11.** Performance improvement after CenterNet optimization demonstration.

After optimization, the overall processing accuracy of CenterNet improved to a certain extent, and it could effectively identify cracks in construction concrete with a shorter training time. The actual detection effect is shown in Figure 12, where the red box represents the detection crack prompts, and the number represents the detection number.

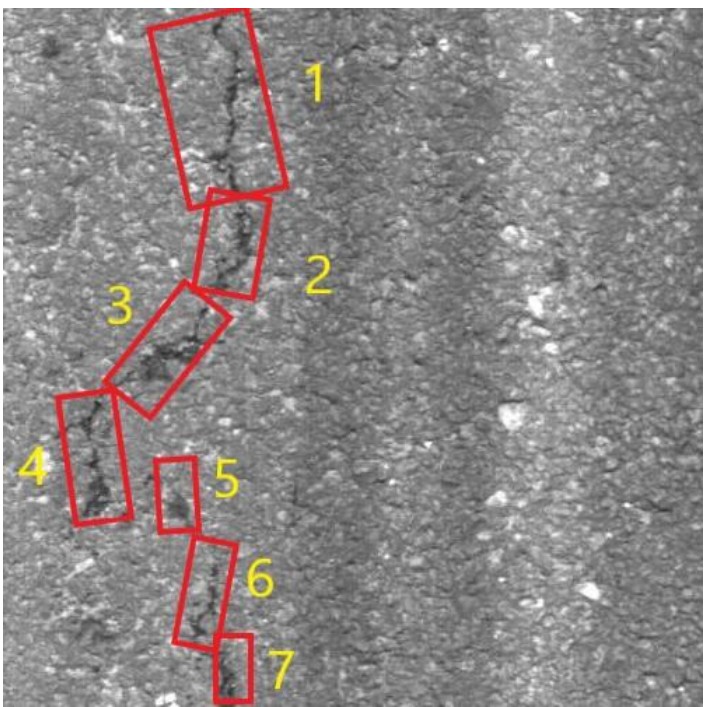

**Figure 12.** Actual detection.

As a classic anchor-free model in the field of computer vision, the CenterNet model has a wide range of applications and optimization in various disciplines. Table 5 shows the comparison between the AP, recall, and $F_1$ score after optimizing the CenterNet model. It is not difficult to find the improvement in the detection effect of the optimization scheme proposed in this paper through comparison.

**Table 5.** Comparison before and after optimization.

| Network | AP/% | R/% | $F_1$ |
|---|---|---|---|
| CenterNet | 75.1 | 50.4 | 53.9 |
| CenterNet-IOU | 77.2 | 51.7 | 57.6 |
| CenterNet-CBAM | 82.3 | 56.2 | 67.2 |
| CenterNet-FS | 85.2 | 53.1 | 60.9 |
| CenterNet-CBAM-FS-IOU | 90.5 | 82.8 | 81.5 |

In order to demonstrate the effectiveness of the proposed method more clearly, it was compared with common crack detection algorithms under the same experimental conditions. The processing results of different crack detection algorithms are shown in Table 6, and the actual detection process is shown in Figure 13. The experimental results show that the method proposed in this paper is superior to other methods, the processing effect of Mask-RCNN has the smallest gap with the method proposed in this paper, and the data processing results are basically close. Through comparison, it can be seen that the method proposed in this article outperforms other detection algorithms in terms of AP, recall, and $F_1$.

**Table 6.** Comparison of different algorithms.

| Network | AP/% | R/% | $F_1$ |
|---|---|---|---|
| YoloV3 | 42.1 | 41.8 | 36.2 |
| YoloV5 | 46.8 | 42.5 | 38.4 |
| YoloV7 | 63.2 | 51.6 | 63.1 |
| Mask-RCNN | 87.9 | 74.3 | 79.8 |
| CenterNet-CBAM-FS-IOU | 90.5 | 82.8 | 81.5 |

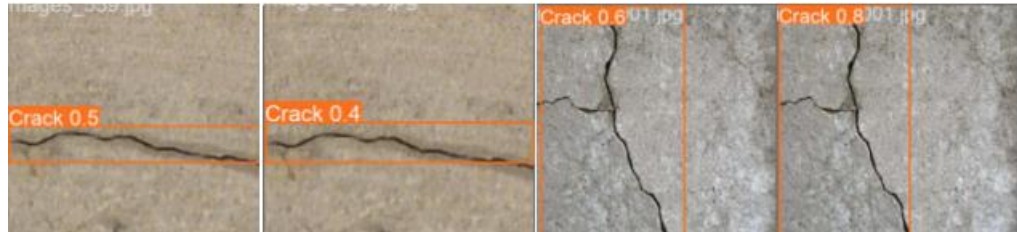

**Figure 13.** Display of different algorithm detection processes.

## 5. Conclusions

Based on the original network of CenterNet, a detailed algorithm model optimization experiment was carried out for the problem of concrete crack detection in construction engineering using pictures of concrete cracks taken by drones, including the addition of a double-attention mechanism, introduction of a feature selection module, and optimization of the loss function.

The experimental results show that the FPS of the improved CenterNet model is reduced by 123.7, the memory is increased by 62 MB, FLOPs are increased by 3.81, and AP is increased by 0.154. The proposed method for detecting cracks in construction projects based on the improved CenterNet network has good robustness and accuracy for the processed datasets and has the potential to be applied for target detection and in recognition methods in relevant practical scenarios.

**Author Contributions:** F.Z.: Finished the paper check work; H.K.: Completed the key data collection, model construction, data processing and paper framework construction of the experiment, and the main writing of the paper; S.G.: Assisted modeling and experimental data acquisition; Q.X.: Performed data collection, model construction, and paper polishing and made great contributions to data processing. All authors have read and agreed to the published version of the manuscript.

**Funding:** This study received no external funding.

**Institutional Review Board Statement:** Not applicable.

**Informed Consent Statement:** Not applicable.

**Data Availability Statement:** Data are contained within the article.

**Acknowledgments:** We offer thanks to Beijing University of Civil Engineering and Architecture for its support.

**Conflicts of Interest:** The authors declare no conflicts of interest.

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
