# Peer review of "Crack Detection of Concrete Based on Improved CenterNet Model"

_applsci, doi:10.3390/app14062527_

Round 1

Reviewer 1 Report

Comments and Suggestions for Authors

The manuscript proposes an improved version of CeneterNet on the concrete crack detection problem. I see some incremental novelty in this work. however, there are some major and minor concerns that should be addressed.

1- The abstract needs to be modified to be more clear. For example, it is not clear whether the AP is increased by 0.154 or 0.544%, the same goes to FLOPs and FPS.

2- The last portion of paragraph 1 in the introduction is poorly written, and it was unclear to the reviewer.

3- The introduction section and literature are insufficient. The authors are suggested to incorporate more literature about deep learning and machine learning techniques on the subject of concrete crack detection. Also, a discussion on any work related to drone-based crack detection and the previous work on real-time crack detection using deep learning and drones. Discussion on the infrared and thermal imaging for crack detection needs to be improved, as well.

4- Introduction section: Please use the full term for C-V.

5- Section 2 (Review on CenterNet): Are the figures in this section borrowed from somewhere? If so, the authors need to reference them. The quality of all the equations in Sections 2 and 3 is very poor (seems images rather than being defined in an equation environment). Please also use equation environment for the terms defined in lines 101-103.

6- Looking at Section 4 (Experimental and Result Analysis), the authors failed to provide information on the dataset. Is borrowed from a repository? If so, what drone and with what payload the data was collected? They need to clearly describe the image sizes, dataset etc. If the data was collected and created by the authors, are they willing to release the dataset an make it public? 

7- Sub-Section 4.4: Did the authors run the experiments with any other optimizer such as Adam, Nadam, etc? 

8- Table 3 needs to be reformatted.

9- More importantly, the authors are highly recommended to compare their results with some variations of Mask-RCNN or Yolo here.

Comments on the Quality of English Language

The authors are suggested to proof-read the paper.

Reviewer 2 Report

Comments and Suggestions for Authors

Dear authors,

I recommend a major revision of the manuscript. My comments and questions are as following:

- The variables in the equations have not been introduced well, e.g., equations 1-4.

- In figure 10 (right), there is still a missing crack out of the blue box, which indicates the deficiency of the proposed algorithm in this paper. Please explain it.

- In figure 12, the cracks are not clear and therefore, it is not clear what have been detected.

- Please provide the results for the benchmark data sets together with accuracy, recall, F1-score, precision and IoU metrices.

Comments on the Quality of English Language

Minor editing of English language is required.

Round 2

Reviewer 1 Report

Comments and Suggestions for Authors

The authors modified the manuscript based on "some" of my previous comments. Still some comments need to be addressed.

The literature review still can be improved and become more comprehensive mainly when it comes to crack detection using this dataset.

The authors are highly recommended to add references to the caption of borrowed figures (both in the body of the manuscript and figure captions)

More importantly, the authors are highly recommended to compare their results with some variations of Mask-RCNN or Yolo here. The authors stated that the simulations have not be accomplished yet due to the time constraints on the revision submission. I highly recommend adding these comparison results with statements of comparisons and details. 

Author Response

Please see the attachment.Thank you for your advice.

Round 3

Reviewer 1 Report

Comments and Suggestions for Authors

Thanks for revising the manuscript and addressing most of those. Two comments still need to be addressed.

1) Simply change the caption in Figure 1 by "Figure 1.Schematic diagram of CenterNet algorithm mode[30].  (i.e., adding the reference at the end of the caption.

2) Highly recommend setting details of your YOLO variations and the comparison results. It is not clear to the reviewer why YOLO worked poorly, maybe due to not rigorous tuning of the parameters. At least some clarification there is recommended.
